# Variable Proportions of Phylogenetic Clustering and Low Levels of Antiviral Drug Resistance among the Major HBV Sub-Genotypes in the Middle East and North Africa

**DOI:** 10.3390/pathogens10101333

**Published:** 2021-10-15

**Authors:** Rabaa Y. Athamneh, Ayşe Arıkan, Murat Sayan, Azmi Mahafzah, Malik Sallam

**Affiliations:** 1Department of Medical Microbiology and Clinical Microbiology, Faculty of Medicine, Near East University, Nicosia 99138, Cyprus; rabaa_athamneh@yahoo.com (R.Y.A.); aysearikancy@yahoo.com (A.A.); 2DESAM, Near East University, Nicosia 99138, Cyprus; sayanmurat@hotmail.com; 3Clinical Laboratory, PCR Unit, Faculty of Medicine, Kocaeli University, İzmit 41380, Turkey; 4Department of Pathology, Microbiology and Forensic Medicine, School of Medicine, the University of Jordan, Amman 11942, Jordan; mahafzaa@ju.edu.jo; 5Department of Clinical Laboratories and Forensic Medicine, Jordan University Hospital, Amman 11942, Jordan; 6Department of Translational Medicine, Faculty of Medicine, Lund University, 22184 Malmö, Sweden

**Keywords:** infectious disease, phylogenetics, evolution, subgenotypes, Morocco, Oman, Sudan, Somalia, UAE, Syria

## Abstract

Hepatitis B virus (HBV) infection remains a major public health threat in the Middle East and North Africa (MENA). Phylogenetic analysis of HBV can be helpful to study the putative transmission links and patterns of inter-country spread of the virus. The objectives of the current study were to analyze the HBV genotype/sub-genotype (SGT) distribution, reverse transcriptase (*RT*), and surface (*S*) gene mutations and to investigate the domestic transmission of HBV in the MENA. All HBV molecular sequences collected in the MENA were retrieved from GenBank as of 30 April 2021. Determination of genotypes/SGT, *RT,* and *S* mutations were based on the Geno2pheno (hbv) 2.0 online tool. For the most prevalent HBV SGTs, maximum likelihood phylogenetic analysis was conducted to identify the putative phylogenetic clusters, with approximate Shimodaira–Hasegawa-like likelihood ratio test values ≥ 0.90, and genetic distance cut-off values ≤ 0.025 substitutions/site as implemented in Cluster Picker. The total number of HBV sequences used for genotype/SGT determination was 4352 that represented a total of 20 MENA countries, with a majority from Iran (*n *= 2103, 48.3%), Saudi Arabia (*n *= 503, 11.6%), Tunisia (*n *= 395, 9.1%), and Turkey (*n *= 267, 6.1%). Genotype D dominated infections in the MENA (86.6%), followed by genotype A (4.1%), with SGT D1 as the most common in 14 MENA countries and SGT D7 dominance in the Maghreb. The highest prevalence of antiviral drug resistance was observed against lamivudine (4.5%) and telbivudine (4.3%). The proportion of domestic phylogenetic clustering was the highest for SGT D7 (61.9%), followed by SGT D2 (28.2%) and genotype E (25.7%). The largest fraction of domestic clusters with evidence of inter-country spread within the MENA was seen in SGT D7 (81.3%). Small networks (containing 3-14 sequences) dominated among domestic phylogenetic clusters. Specific patterns of HBV genetic diversity were seen in the MENA with SGT D1 dominance in the Levant, Iran, and Turkey; SGT D7 dominance in the Maghreb; and extensive diversity in Saudi Arabia and Egypt. A low prevalence of lamivudine, telbivudine, and entecavir drug resistance was observed in the region, with almost an absence of resistance to tenofovir and adefovir. Variable proportions of phylogenetic clustering indicated prominent domestic transmission of SGT D7 (particularly in the Maghreb) and relatively high levels of virus mobility in SGT D1.

## 1. Introduction

Chronic hepatitis B (CHB) is one of the major global health care issues, with more than a quarter-billion people chronically infected by hepatitis B virus (HBV), resulting in more than 800,000 deaths as of 2015 [1,2]. Despite the availability of an effective HBV vaccine and the accelerated advent of antiviral therapies to control CHB, HBV has been suggested to cause almost half of all deaths globally from viral hepatitis, half of all deaths from hepatocellular carcinoma (HCC), and a third of all mortalities from hepatic cirrhosis [3,4,5].

The substantial global burden of CHB and its complications mandates continuous and comprehensive research involving its epidemiology, the spread of antiviral drug resistance, and the emergence of vaccine-escape mutations in the virus [6,7,8]. This can help in refining knowledge regarding HBV and CHB. Ultimately, this can aid in achieving the World Health Organization (WHO) goal of eliminating hepatitis B as a public health threat (which entails a 90% reduction in new chronic infections and a 65% reduction in mortality) [9,10].

The genome of HBV is a circular, partially double-stranded DNA with four overlapping open reading frames (ORFs) and a length of approximately 3.2 kilobases (kb) [11]. These four ORFs encode the polymerase (P), surface (S), core (C), and X proteins [11]. This virus is unique among human DNA viruses due to its reverse-transcriptase that lacks a proofreading activity [12,13]. The consequence of such a replication mechanism is a relatively higher evolutionary rate compared to other DNA viruses [14]. The extensive genetic diversity of HBV, besides the rapid development of drug resistance and vaccine escape mutations, are amongst the manifestations of such a rapid evolution [15].

On the other hand, the presence of overlapping ORFs in the viral genome constrains the evolutionary rate of HBV since a synonymous mutation in one ORF can result in a non-synonymous mutation in another ORF [16,17,18]. The relatively swift evolution of HBV has resulted in its diversification into several genotypes (A–J), which are further classified into multiple sub-genotypes (SGTs) assigned to Arabic numbers [17,19,20,21,22]. Genotype assignment is based on pairwise inter-genotype genetic distance > 8.0%, while SGTs are grouped together based on pairwise genetic distances of 4.0–8.0% [19]. The gold-standard method for HBV genotyping is phylogenetic analysis, preferably of full-genomes; however, analysis of the *S* gene region that overlaps with the *P* region can be sufficient for accurate genotype determination [19].

The epidemiologic investigation of HBV infection reveals considerable variability in terms of prevalence, genotype distribution, and transmission patterns. Based on rates of the chronic carriage of hepatitis B surface antigen (HBsAg), countries can be classified as countries with hyperendemicity (HBsAg prevalence of > 8.0%), intermediate endemicity (HBsAg prevalence of 2.0–8.0%), and low endemicity with HBsAg prevalence < 2.0% [23,24,25].

Five HBV genotypes cause 96% of CHB cases worldwide, with genotype C as the most common (26%), followed by genotype D (22%), genotype E (18%), genotype A (17%), and genotype B (14%) [26]. Genotype distribution varies in different regions, with genotype D predominating in the Mediterranean region, while genotypes C and B are mostly reported in South East Asia, China, and Japan, and genotype A is most commonly found in the United States and Northern Europe [26,27].

The potential clinical value of HBV genotyping has gained much attention recently, with examples including possible association with more severe disease among the patients infected with genotype C compared to those with genotype B infection, and another example pointing to a possible earlier occurrence of HCC among patients infected with genotype D compared to those with genotype A infection [8,28,29,30,31]. In contrast, other studies found no such link [32].

Transmission of HBV occurs mainly through the parenteral route, besides horizontal, sexual, or vertical routes [33,34,35]. A few studies reported potential links between certain HBV genotypes and possible routes of transmission: the preferred route for genotype A was sexual transmission, while genotype D was mainly transmitted by blood transfusion and transplantation [32]. 

The Middle East and North Africa (MENA) represent a region with shared demographic, cultural, and economic attributes [36]. The study of virus transmission in the region as a single unit can be invaluable in the epidemiologic investigation of virus spread [37,38,39,40,41,42]. The MENA region can be classified among the regions with intermediate to high endemicity of CHB [41,43]. In addition, CHB was reported as an important cause of HCC in the region [44]. Genotype D was found to dominate infections in the MENA region from various countries in different time periods [45,46,47,48,49,50,51,52,53]. This appears as an expected distribution considering that the MENA region is the putative epicenter for genotype D of the virus, with widespread distribution of its SGTs in the region [54,55,56]. Despite the recent reports indicating that the coverage of the HBV vaccine has increased to exceed 80% for the third dose, the low birth dose coverage remains a challenge in the MENA region [57].

The objectives of the current study were: (1) To characterize the genetic diversity of HBV in the MENA region; (2) To estimate the prevalence of the *RT* and *S* genes’ mutations in the region; (3) To assess the proportions of phylogenetic clustering in the MENA region as indicators of domestic transmission of the virus.

## 2. Materials and Methods

### 2.1. Study Design and Sequence Inclusion Criteria

Using the search tool in GenBank (the National Institutes of Health (NIH)) genetic sequence database (www.ncbi.nlm.nih.gov/genbank/ accessed on 30 April 2021), a search was conducted for all HBV sequences that were collected from the following countries/territories of the MENA region: Algeria, Bahrain, Cyprus/Northern Cyprus, Egypt, Iran, Iraq, Jordan, Kingdom of Saudi Arabia (KSA), Kuwait, Lebanon, Libya, Mauritania, Morocco, Oman, Palestine, Qatar, Somalia, Sudan, Syria, Tunisia, Turkey, United Arab Emirates (UAE), and Yemen [58]. The search was completed on 30 April 2021, and the following sequence metadata were also retrieved (if available): year and country of sequence collection, genotype/SGT data, and sequence length.

### 2.2. Determination of HBV Genotype Distribution, Antiviral Resistance and S Gene Mutations

In order to investigate the HBV genotype distribution in the MENA, the final dataset was compiled using the following exclusion criteria: (1) Country of collection other than MENA countries described in the section above; (2) Sequence length of less than 300 base pairs (bp, based on the assumption that shorter sequences can yield inaccurate sub-genotyping results); or (3) Sequences that did not span a part of the *P/S* ORFs. The determination of HBV genotypes and SGTs, antiviral drug resistance, and *S* gene mutations were conducted using the Geno2pheno (HBV) online tool and compared with GenBank metadata [59]. Results for sub-genotyping were mostly concordant except for SGT D7, with sequences assigned to SGT D4 in Geno2pheno (HBV) online tool. These sequences were considered as SGT D7 based on original nomenclatures used by the original GenBank submitters/authors.

### 2.3. Analysis of HBV Domestic Transmission in the MENA Region

In order to investigate the domestic transmission of the most prevalent HBV SGTs in the MENA (SGTs with > 10 MENA sequences), the final datasets were compiled according to the following inclusion criteria: (1) Molecular sequences covering the genomic region that was selected for the final analysis, which spanned part of the *P* and *S* genes (positions 216–755 in relation to HBV reference genome with GenBank accession NC_003977); and (2) Final SGT dataset with at least 10 MENA sequences. The final number of molecular HBV sequences that were considered for phylogenetic clustering analysis was 2384.

Phylogenetic inference of possible transmission links among the MENA HBV sequences was performed using the maximum likelihood (ML) approach as implemented in PhyML 3.0 [60]. A search for similar HBV GenBank sequences was performed using the BLAST tool, looking for the ten most similar sequences, which were included for final ML analysis together with the MENA sequences for each SGT [61]. The global sequences without known country of origin and those with stop codons were excluded from the final analysis.

The criteria to identify phylogenetic clusters indicating putative epidemiologic linkages were: (1) Internal branch support values ≥ 0.90 using approximate Shimodaira–Hasegawa-like likelihood ratio test (aLRT-SH); (2) An ad hoc mean intra-cluster genetic distance of ≤ 2.5%; (3) More than or equal to 75.0% sequences collected within the MENA countries/territories [62,63]. Selection of the genetic distance was in light of HIV phylogenetic cluster analysis basis together with one study investigating familial clustering of HBV [64,65]. Transmission cluster analysis was performed using Cluster Picker 1.2 tool to identify the putative monophyletic clades in the MENA region [66]. Based on the cluster size, each cluster was classified as dyads (two sequences), small networks (three-14 sequences), or large networks (≥15 sequences) [37,67,68].

### 2.4. Statistical Analysis

Analysis was conducted with IBM SPSS Statistics for Windows, Version 22.0. Armonk, NY: IBM Corp. The possible associations between categorical variables were evaluated using the chi-square (χ^2^) test. The statistical significance was considered for *p* < 0.010.

## 3. Results

### 3.1. Characteristics of the MENA HBV Molecular Dataset

The total number of sequences that spanned part of the *P* gene and were >300 base pairs in length was 4352, which formed the basis for the final genotyping of HBV in the MENA (Figure 1). Three-quarters of all sequences came from four MENA countries: Iran (*n *= 2103, 48.3%), KSA (*n *= 503, 11.6%), Tunisia (*n *= 395, 9.1%), and Turkey (*n* = 267, 6.1%, Figure 2). The years of sequence collection ranged from 2000 to 2019, with a total of 1329 sequences that lacked the dates of collection. 

### 3.2. Genotype D Dominated HBV Infections in the MENA despite the Detection of an Extensive Genetic Diversity

In the 20 MENA countries/territories with HBV sequences eligible for HBV genotype analysis, genotype D was the most common in 19 countries. Somalia was the only country with genotype A as the most common HBV genotype (Figure 3).

Egypt, KSA, and Tunisia were the only MENA region countries where the eight HBV genotypes (A–H) were detected.

Iran, Iraq, and Turkey, together with countries of the Levant region (Jordan, Lebanon, Palestine, and Syria), were dominated by genotype D (Iran: 99.9%; Turkey: 92.9%; Iraq: 89.5%; Jordan, Lebanon, and Syria: 100.0%; and Palestine: 93.2%).

Countries of the Middle East displayed a significantly higher prevalence of genotype D vs. all other genotypes grouped together compared to countries of MENA African countries (3195/3484 (91.7%) vs. 572/868 (65.9%); *p* < 0.001, χ^2^ test, Figure 4).

### 3.3. HBV Sub-Genotype Distribution in the MENA Region

For genotypes A–D, sub-genotyping results were available for 4340 sequences with 12 sequences identified to genotype level only (genotype A from Morocco, *n *= 1; genotype D from Iran, *n *= 9; and genotype F from KSA, *n *= 2). Overall, the most common HBV SGT observed in the MENA was D1 (*n *= 3021/4340, 69.6%), followed by D7 (*n *= 508, 11.7%), D2 (*n *= 156, 3.6%), A1 (*n *= 141, 3.2%), genotype E (*n *= 112, 2.6%), genotype G (*n *= 74, 1.7%), and D3 (*n *= 73, 1.7%).

The most extensive genetic diversity of HBV was observed in KSA, where 16 different genotypes/SGTs were found, followed by Egypt (15 different genotypes/SGTs), Tunisia (12 different genotypes/SGTs), and Turkey (10 different genotypes/SGTs, Table 1). Jordan was the only country with the exclusive presence of a single SGT (D1, Table 1).

Sub-genotype D1 was the most common genetic variant of HBV in 14 out of the 20 MENA region countries/territories. Sub-genotype D7 was the most common in the Maghreb countries (Morocco, Algeria, and Tunisia), while in Sudan, genotype E dominated, and in Somalia, SGT A1 was the most common (Figure 5).

### 3.4. Antiviral Drug Resistance and S Gene Mutations in the MENA HBV Sequences

For antiviral drug resistance prediction, the MENA HBV sequences that were assigned to the year of the collection were only used (*n *= 3023), excluding the sequences that lacked such data (*n* = 1329). The estimates for each drug were used for susceptible vs. resistant/partly resistant/limited susceptibility categories, while the rest were excluded (unknown, compensatory). The overall prevalence of resistance to the five antiviral drugs is shown in (Table 2).

No significant temporal changes in resistance among HBV MENA sequences were observed; however, spatial differences were detected with a higher percentage of resistance to lamivudine, entecavir, and telbivudine in the Middle East compared to North Africa (Table 3). The complete list of mutations detected in each of the five HBV antiviral classes is summarized in (Table 4).

For the *S* gene, the most frequent mutations detected are summarized in (Figure 6). The most common *S* gene mutations included: 143L (*n *= 373), 144A (*n *= 327), 126I (*n *= 323), and 141R (*n *= 302).

### 3.5. Variable Proportions of Putative Domestic HBV Transmission in the MENA Region for Different SGTs

The final number of molecular HBV sequences that were considered for phylogenetic clustering analysis was 2375, based on the genomic region that was selected for final analysis, which spanned part of the *P* and *S* genes (positions 216–755 in relation to HBV reference genome with GenBank accession no. NC_003977). The selection of this region was based on an initial analysis of the HBV genetic region with the highest number of MENA sequences (Figure 7). Sub-genotypes that had at least 10 sequences that were collected in the MENA region included in descending order: D1 (*n *= 1777), D7 (*n *= 268), A1 (*n *= 102), D2 (*n *= 78), E (*n *= 73), D3 (*n *= 48), and A2 (*n *= 23). The excluded SGTs from transmission cluster analyses included: C2 (*n *= 3), B1 (*n *= 1), B4 (*n *= 1), and C1 (*n *= 1).

For SGT D1, the total number of MENA sequences enclosed within domestic phylogenetic clusters was 267/1777 (15.0%). These domestic clusters comprised 74 sequences in dyads (27.7%), small networks (*n *= 167, 62.5%), and a single large network (26, 9.7%). For SGT D7, the total number of MENA sequences enclosed within domestic phylogenetic clusters was 166/268 (61.9%). These domestic clusters comprised 18 sequences in dyads (10.3%), small networks (*n *= 83, 47.7%), and a single large network (73, 42.0%). For SGT D2, the total number of MENA sequences enclosed within domestic phylogenetic clusters was 22/78 (28.2%). The clusters comprised five dyads and two small networks, each with five MENA sequences. For genotype E, the total number of MENA sequences enclosed within domestic phylogenetic clusters was 18/73 (25.7%). These domestic clusters were divided into a single dyad (*n *= 2, 11.1%), a single small network (*n* = 3, 16.7%), and a large network (*n *= 13, 72.2%). A single small network was noticed for SGT D3 with an overall proportion of clustering of 4/48 (8.3%). For SGTs A1 and A2, no domestic clusters were identified.

The largest fraction of sequences from different MENA countries were seen in SGT D7 (135/166, 81.3%). Inter-country spread of SGT D1 was seen in clusters containing a total of 66 sequences (24.7%), while for SGT D2, inter-country spread comprised 5/22 (22.7%), and it was 3/18 (16.7%) for genotype E. Out of the total 109 MENA clusters identified in this study, 30 clusters contained sequences from two or more MENA countries. A fraction of the transmission clusters with MENA sequences collected in two or more countries originated from neighboring countries: Iran and Oman (D1 dyad, D7 large network), Egypt and KSA (D1 small network), Egypt and Sudan (E small network), Iran and Turkey (D1 two small networks), and Algeria, Morocco and Tunisia (D7 two small networks, Table 5). All ML trees used to infer the putative transmission clusters in the MENA region are provided in (Appendix A).

## 4. Discussion

The MENA region can be viewed as a region with a relatively high burden of hepatitis B [41,69,70]. In this study, we investigated HBV genetic diversity, antiviral drug resistance, and transmission clustering patterns in the MENA region using a phylogenetic-based approach. Countries of the region can be classified as intermediate to highly endemic in relation to the prevalence of CHB [27,41,71]. The study of viral transmission in a region with a common culture, economic difficulties, belief systems, and behaviors can be valuable to reveal the dynamics of virus spread [37,72,73,74]. In addition, the study of HBV genotype distribution is valuable considering its potential association with the progression of liver disease, viral load, and viral clearance [75,76]. 

A major result of this study was the observation of the extensive genetic diversity of HBV in the MENA region. Nevertheless, genotype D dominated infections in the majority of the MENA countries, particularly in the Middle East. Additionally, the results indicated that genotype D emerged as the most significant genetic variant of the virus from an epidemiologic point of view due to its contribution to the local spread of HBV. This result appears plausible considering the previous evidence of genotype D dominance in the Eastern Mediterranean region [55,77,78]. Genotype D was previously reported to be the most dominant variant of HBV in Iran, Iraq, Syria, Jordan, Lebanon, Palestine, Turkey, and Northern Cyprus [46,50,53,79,80,81]. In Iran, two separate studies by Aghakhani et al. and Pourkarim et al*.* attributed the dominance of genotype D in the country to its geographical location and ethnic background [82,83]. 

The extensive genetic diversity of HBV in the MENA was more pronounced in some countries (KSA and Egypt). Saudi Arabia has a high level of international migration within the MENA region countries [84]. Thus, the large diversity of residents in KSA was reflected by the presence of the eight HBV genotypes in the country. A previous study by Al-Qudari et al. demonstrated such an extensive genetic diversity of HBV in KSA [85]. For Egypt—particularly in Cairo—large urban refugee communities existed for a long time, which might have contributed to the huge genetic diversity of the virus in the country [86]. Saudy et al. suggested that being a destination for many tourists and visitors, particularly from the countries where genotype D is prevalent, can explain the considerable genetic diversity and dominance of genotype D in Egypt [87].

Likewise, Sumer and Sayan suggested that the HBV genetic diversity seen in Northern Cyprus can be related to the relatively large fraction of students and workers present there, with similar evidence by Arıkan et al*.* [53,88]. An important point to be considered in areas with extensive genetic diversity of the virus is the higher possibility of recombination, which might yield novel viral variants, that may impact the prophylaxis, diagnosis, and treatment of the disease [89,90].

Another interesting observation was the dominance of SGT D7 in the Arab Maghreb (Northwest Africa). The sustained presence of SGT D7 in the Maghreb was linked to intra-familial transmission in early childhood [91]. The dominance of SGT D7 in the Maghreb was previously reported by several original and review papers and is consistent with the results of this study [49,91,92,93,94].

The early characterization of SGT D7 was reported in Tunisia by Meldal et al., and the evolution of this SGT in Tunisia was further dissected by Ciccozzi et al., who described its peculiar distribution geographically [91,94]. The latter study demonstrated a recent origin of SGT D7 dating back to the 1950s, with an exponential increase in infections through two main routes, namely, familial transmission and the unsafe use of needles [91]. The peculiarity of SGT D7 in this study extended to involve the classification with the Geno2pheno (HBV) online tool, where this SGT was initially classified as SGT D4. The consistent discrepancy between the GenBank sequence metadata and genotyping results, besides the lack of SGT D4 retrieval using the BLAST tool with D7 as the query sequences, demonstrated its genuine classification as D7 rather than D4, which is found mainly in Oceania rather than the MENA region [55]. A possible explanation of SGT D7’s initial misclassification as SGT D4 is the shared recent common ancestor for both SGTs D4 and D7, with its possible divergence from SGT D5 that took place in the Maghreb as suggested by Ciccozzi et al. [91]. This result points to the importance of conducting phylogenetic classification using full genomes to achieve reliable conclusions about the genetic diversity of the virus and the continuous need for revising the classification for HBV, which is characterized by a swift evolution among human DNA viruses.

Other studies from both Algeria and Morocco add further evidence to the previous hypothesis of SGT D7 origin in the Maghreb [49,95]. Collaborative efforts are required in the region to help in better characterization of the HBV epidemic in the Maghreb, which in turn can help in the implementation of well-informed preventive public health measures to reduce the burden of CHB in the region [92].

Despite its global distribution, genotype D predominates around the Mediterranean and in the Middle East [15,26]. The clinical significance of genotype D stems from its possible association with poor prognosis, besides its higher correlation with acute liver failure [15,75,96,97,98]. Previous studies investigating the origin of SGT D1 using the phylogeographic approach pointed to possible origins of this sub-genotype in the region (Syria and Turkey); thus, the predominance of SGT D1 in the Levant appears as an expected outcome [99,100,101]. The dominance of genotype E in Sudan contradicts a previous report with a small sample size in the country, which showed that genotype D was the most common, followed by genotype E [102]. However, the higher number of molecular sequences analyzed in the current study might point to the genuine dominance of genotype E in Sudan, which appears as a possible outcome considering the high prevalence of this genotype in Sub-Saharan Africa [103].

The proportion of phylogenetic clustering besides inter-country mixing of HBV lineages was the highest in Maghreb. This was higher than the proportions previously reported for HIV in Europe and the MENA region [37,73]. Taken together, the results of phylogenetic cluster analysis can give clues to very high levels of viral mobility for the following: SGTs A1 and A2; high levels for SGTs D1, D3, and genotype E; and high levels of domestic spread with inter-country mixing of SGT D7 viral lineages.

A previous phylogenetic suggestion of HBV movement in the MENA region was reported by Garmiri et al., where Iranian and Egyptian strains clustered together despite the lack of high statistical support using bootstrapping method [104]. The study by Pourkarim et al. that reported the clustering of Iranian HBV strains with isolates from Turkey, Syria, and Lebanon was in line with our results [83].

Another recent study from Jordan demonstrated evidence of possible inter-country spread of HBV in the MENA region as inferred through the observation of intermingling of HBV sequences from Iran, Turkey, Syria, and KSA with Jordanian sequences [50]. Nevertheless, the previous study by Ababneh et al. reported a higher proportion of SGT D1 clustering (30%) compared to the results of this study (15%) [50]. Possible explanations for this discrepancy can be the adoption of a stricter definition of statistical support for the internal nodes in defining the monophyletic clades in this study besides the examination of a larger dataset from a region rather than a single country. 

Considering the evidence of domestic viral spread in the region, halting dispersal of HBV requires collaborative efforts and more comprehensive epidemiologic studies to identify possible risk factors of infection and proper vaccine coverage, which can be used to guide well-informed focused preventive measures [105].

Regarding the most common *S* gene mutations detected in this study, such mutations can have a notable impact on the biologic behavior of the virus as follows: changing antigenicity of the surface antigen, which may affect the specificity of monoclonal antibodies binding to this antigen [106,107]; reduction in the antigenicity of the surface protein of HBV with subsequent reactivation resulting in occult HBV infection [108]; the lower possibility of detection resulting in a higher risk of transmission in association with occult hepatitis B infection [108]; and possible reduction in the ability to detect HBV surface antibodies in immunoassays [109,110]. Such mutations should be monitored closely considering the potential survival advantage of such immune-escape mutants [111]. 

For the polymerase gene mutations, the overall prevalence of those mutations conferring antiviral drug resistance was low (mainly against lamivudine and telbivudine). Nevertheless, continuous monitoring of such mutations is recommended to prevent the selection of such mutants, which would hinder the management of CHB [112]. This result should be interpreted with extreme caution due to the lack of data for the treatment status of the individuals from whom the molecular HBV sequences were retrieved and analyzed in this study. The previous reports of the rapid emergence of resistance to lamivudine also shed doubt over this particular result [113,114,115]. Thus, more studies from various countries of the region are recommended to confirm or disprove such an uncertain result. Such research is particularly recommended in the Middle East considering the significantly higher prevalence of resistance to lamivudine, entecavir, and telbivudine.

The RT amino acid substitutions 204I/V were among the most frequent RT substitutions detected in the study. Such amino acid substitutions were reported previously in Iran and Jordan, even among treatment-naïve individuals [50,116,117]. The amino acid substitutions 204I/V can be considered among the signature resistance mutations to lamivudine with cross-resistance to telbivudine [118,119]. The decreased response to lamivudine associated with the selection of drug-resistant mutants was reported in Egypt and Iran and should be considered carefully since this antiviral is considered a cost-effective treatment option widely used in the MENA region [71,120,121].

The current study had several limitations as follows: first, a pre-requisite for the determination of HBV SGTs is the analysis of full-length genomes, which was not available in this study. Thus, slight differences in SGT assignments should be expected [122]. Second, a limitation that should be considered in any prospective research investigating HBV epidemiology is the need to consider the probable mode of transmission and serologic profiles among the infected patients, besides the treatment status, all of which were not available in the majority of sequences utilized in this study [123]. Third, the use of short genomic regions and sparse sampling were unavoidable in this study as well, which could have influenced the estimates of HBV clustering. Fourth, the possibility of recombination in the analyzed sequences was not ruled out. Finally, a few countries from the MENA lacked HBV molecular sequences in GenBank (Bahrain, Djibouti, Mauritania, and Qatar), besides the unequal distribution of the number of HBV sequences in this study with a predominance of sequences from Iran, Saudi Arabia, and Tunisia, which could result in biased results.

## 5. Conclusions

The following patterns of HBV genotype distribution were observed in the MENA: SGT D1 dominance in the Levant, Iran, and Turkey, SGT D7 dominance in the Maghreb, genotype E dominance in Sudan, and SGT A1 dominance in Somalia. In addition, an extensive genetic diversity of HBV was seen in Saudi Arabia and Egypt. Low prevalence of lamivudine, telbivudine, and entecavir drug resistance in the MENA was found in this study, with almost absence of resistance to tenofovir and adefovir. Transmission cluster phylogenetic analysis indicated the variable proportions of phylogenetic clustering. Specifically, the prominent domestic transmission of SGT D7 was observed (particularly in the Maghreb), while lower levels of clustering observed for genotype E and SGTs D1 andD3 suggests higher levels of virus mobility and inter-country spread of the virus in the region. Future epidemiologic studies involving the investigation of risk factors, vaccine coverage, the occurrence of antiviral drug resistance, and vaccine escape mutations are highly recommended to refine the preventive and management strategies in the MENA.

## Figures and Tables

**Figure 1 pathogens-10-01333-f001:**
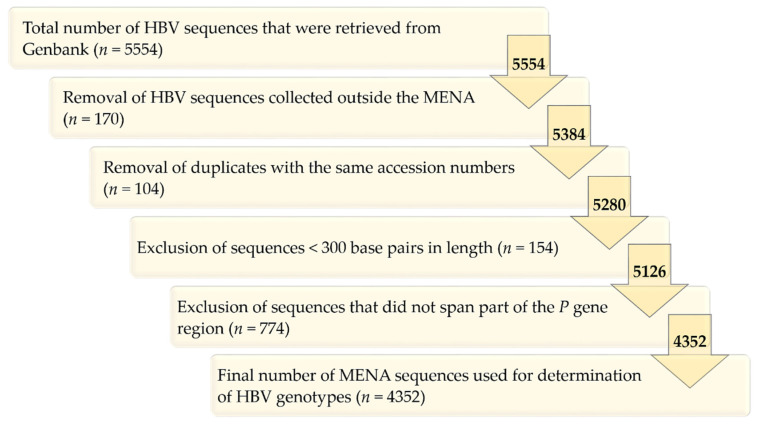
Selection algorithm for the MENA HBV sequences that formed the basis for sub-genotyping. HBV: hepatitis B virus; MENA: Middle East and North Africa; *P*: Polymerase gene region.

**Figure 2 pathogens-10-01333-f002:**
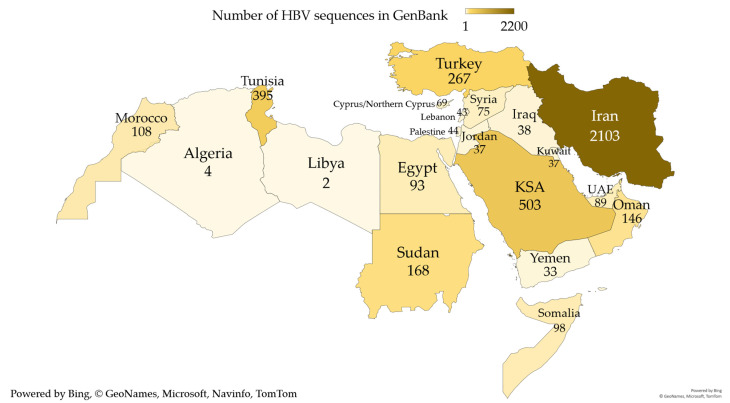
The total number of HBV MENA sequences that were used for sub-genotyping stratified per country. HBV: hepatitis B virus; MENA: Middle East and North Africa. The map only shows the MENA countries with HBV sequences. The darker colors indicate higher number of HBV sequences. The map was generated in Microsoft Excel, powered by Bing, © GeoNames, Microsoft, Navinfo, TomTom, Wikipedia. We are neutral with regard to jurisdictional claims in this map.

**Figure 3 pathogens-10-01333-f003:**
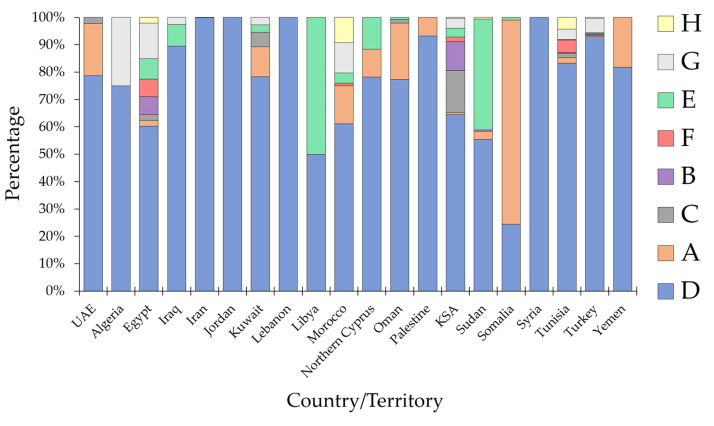
Hepatitis B virus genotype distribution per country in the MENA region. MENA: Middle East and North Africa; KSA: Saudi Arabia; UAE: United Arab Emirates.

**Figure 4 pathogens-10-01333-f004:**
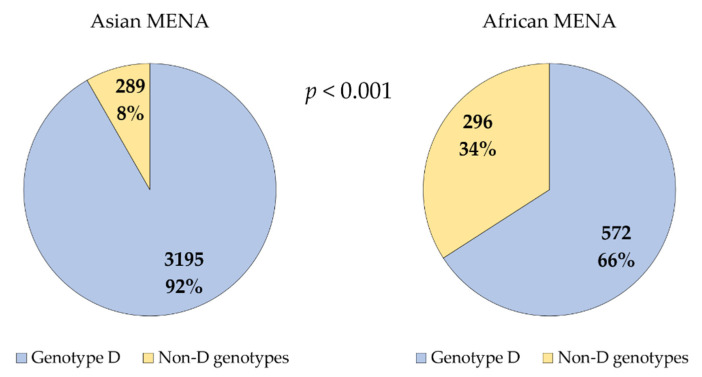
Hepatitis B virus genotype D distribution in Asian vs. African MENA countries. MENA: Middle East and North Africa; p value was calculated using chi-square test.

**Figure 5 pathogens-10-01333-f005:**
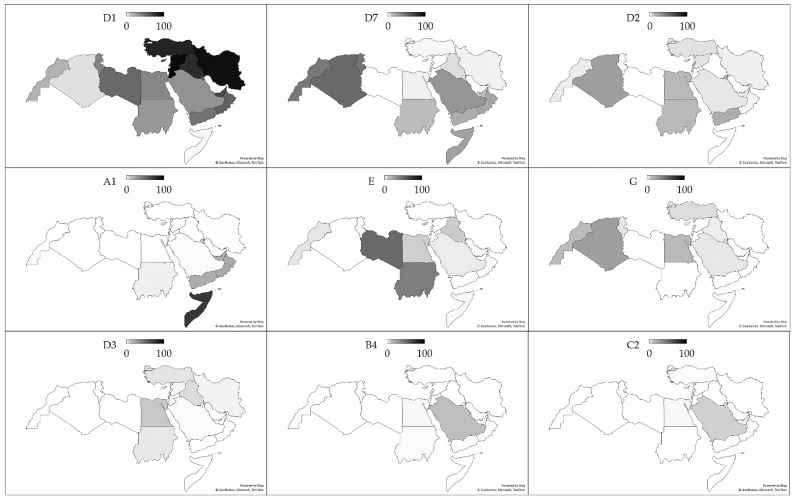
The percentage of hepatitis B virus genotype/sub-genotypes per country in the Middle East and North Africa region. The darker colors indicate higher proportions of each genotype/sub-genotype. The map was generated in Microsoft Excel, powered by Bing, © GeoNames, Microsoft, Navinfo, TomTom, Wikipedia. We are neutral with regard to jurisdictional claims in this map.

**Figure 6 pathogens-10-01333-f006:**
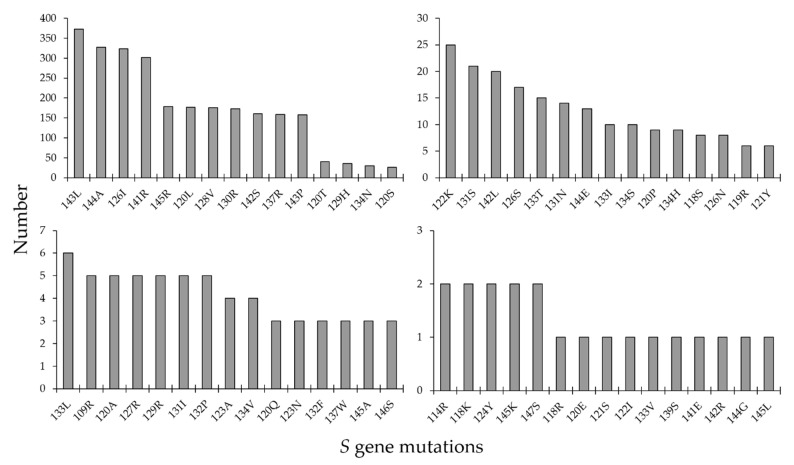
Surface (*S*) gene mutations among hepatitis B virus molecular sequences collected in the Middle East and North Africa region.

**Figure 7 pathogens-10-01333-f007:**
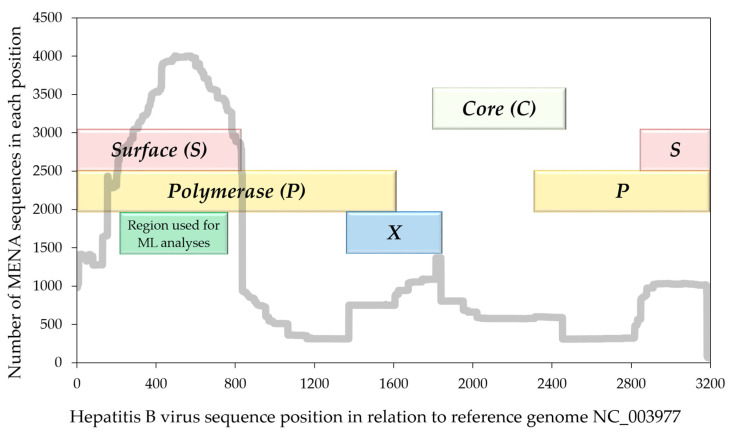
The number of HBV MENA sequences in each genome position in relation to the reference HBV genome. HBV: Hepatitis B virus; MENA: Middle East and North Africa; ML: Maximum likelihood phylogenetic analyses; the reference HBV genome used had the GenBank accession no. NC_003977.

**Table 1 pathogens-10-01333-t001:** Hepatitis B virus genotype/sub-genotype detailed distribution per country/territory of the MENA region.

Country (Total Number of HBV Sequences) ^1^	The Most Common HBV Genotype/SGT ^4^ (*n*, %)	Other SGTs
Algeria (4)	D4 (*n* = 2, 50.0%)	D2 (*n* = 1, 25.0%), G (*n* = 1, 25.0%)
Egypt (93)	D1 (*n* = 34, 36.6%)	D2 (*n* = 12, 12.9%), G (*n* = 12, 12.9%), D3 (*n* = 8, 8.6%), E (*n* = 7, 7.5%), B1 (*n* = 4, 4.3%), F1 (*n* = 3, 3.2%), F2 (*n* = 12, 12.9%), D7 (*n* = 2, 2.2%), A2 (*n* = 2, 2.2%), H (*n* = 2, 2.2%), B2 (*n* = 1, 1.1%), B4 (*n* = 1, 1.1%), C1 (*n* = 1, 1.1%), C2 (*n* = 1, 1.1%)
Iraq (38)	D1 (*n* = 30, 79.0%)	E (*n* = 3, 7.9%), D3 (*n* = 2, 5.3%), D7 (*n* = 2, 5.3%), G (*n* = 1, 2.6%)
Iran (2094)	D1 (*n* = 1958, 93.5%)	D2 (*n* = 52, 2.5%), D7 (*n* = 41, 2.0%), D3 (*n* = 40, 1.9%), A1 (*n* = 2, 0.1%), A2 (*n* = 52, 2.5%)
Jordan (37)	D1 (*n* = 37, 100.0%)	-
Kuwait (37)	D1 (*n* = 23, 62.2%)	D2 (*n* = 6, 16.2%), A1 (*n* = 3, 8.1%), C2 (*n* = 2, 5.4%), A2 (*n* = 1, 2.7%), E (*n* = 1, 2.7%), G (*n* = 1, 2.7%)
Lebanon (43)	D1 (*n* = 36, 83.7%)	D2 (*n* = 7, 16.3%)
Libya (2)	D1 and E (*n* = 1, 50.0%)	-
Morocco (107)	D7 (*n* = 46, 43.0%)	D1 (*n* = 17, 15.9%), A2 (*n* = 14, 13.1%), G (*n* = 12, 11.2%), H (*n* = 10, 9.3%), E (*n* = 4, 3.7%), D2 (*n* = 3, 2.8%), F2 (*n* = 1, 0.9%)
Northern Cyprus (69)	D1 (*n* = 49, 71.0%)	E (*n* = 8, 11.6%), A1 (*n* = 5, 7.2%), D2 (*n* = 4, 5.8%), A2 (*n* = 2, 2.9%), D3 (*n* = 1, 1.4%)
Oman (146)	D1 (*n* = 79, 54.1%)	D7 (*n* = 32, 21.9%), A1 (*n* = 30, 20.6%), D2 (*n* = 2, 1.4%), E (*n* = 1, 0.7%), C1 (*n* = 1, 0.7%), C2 (*n* = 1, 0.7%)
Palestine (44)	D1 (*n* = 39, 88.6%)	A2 (*n* = 3, 6.8%), D2 (*n* = 1, 2.3%), D3 (*n* = 1, 2.3%)
KSA ^2^ (501)	D1 (*n* = 155, 30.9%)	D7 (*n* = 147, 29.3%), B4 (*n* = 51, 10.2%), C2 (*n* = 36, 7.2%), C4 (*n* = 31, 6.2%), D2 (*n* = 19, 3.8%), G (*n* = 18, 3.6%), E (*n* = 16, 3.2%), C1 (*n* = 7, 1.4%), F2 (*n* = 6, 1.2%), D3 (*n* = 3, 0.6%), A1 (*n* = 3, 0.6%), B2 (*n* = 3, 0.6%), C3 (*n* = 3, 0.6%), H (*n* = 2, 0.4%), A2 (*n* = 1, 0.2%)
Sudan (168)	E (*n* = 68, 40.5%)	D1 (*n* = 47, 28.0%), D2 (*n* = 21, 12.5%), D7 (*n* = 19, 11.3%), D3 (*n* = 6, 3.6%), D4 (*n* = 147, 29.3%), A1 (*n* = 4, 2.4%), B4 (*n* = 1, 0.6%), H (*n* = 1, 0.6%), A2 (*n* = 1, 0.6%)
Somalia (98)	A1 (*n* = 72, 73.5%)	D7 (*n* = 23, 23.5%), E (*n* = 1, 1.0%), D1 (*n* = 1, 1.0%), A2 (*n* = 1, 1.0%)
Syria (75)	D1 (*n* = 71, 94.7%)	D2 (*n* = 3, 4.0%), D3 (*n* = 1, 1.3%)
Tunisia (395)	D7 (*n* = 176, 44.6%)	D1 (*n* = 149, 37.7%), F2 (*n* = 18, 4.6%), H (*n* = 17, 4.3%), G (*n* = 15, 3.8%), A2 (*n* = 8, 2.0%), D2 (*n* = 4, 1.0%), C2 (*n* = 3, 0.8%), E (*n* = 1, 0.3%), C4 (*n* = 1, 0.3%), B2 (*n* = 1, 0.3%)
Turkey (267)	D1 (*n* = 222, 83.2%)	G (*n* = 14, 5.2%), D2 (*n* = 12, 4.5%), D3 (*n* = 11, 4.1%), D7 (*n* = 3, 1.1%), C3 (*n* = 2, 0.5%), H (*n* = 1, 0.4%), A2 (*n* = 1, 0.4%), C2 (*n* = 1, 0.4%), E (*n* = 1, 0.4%), B3 (*n* = 1, 0.4%)
UAE ^3^ (89)	D1 (*n* = 58, 65.2%)	A1 (*n* = 16, 18.0%), D7 (*n* = 9, 10.1%), D2 (*n* = 3, 3.4%), C2 (*n* = 2, 2.2%), A2 (*n* = 1, 1.1%)
Yemen (33)	D1 (*n* = 15, 45.5%)	A1 (*n* = 6, 18.2%), D2 (*n* = 6, 18.2%), D7 (*n* = 6, 18.2%)

^1^ HBV: Hepatitis B virus; ^2^ KSA: Saudi Arabia; ^3^ UAE: United Arab Emirates; ^4^ SGT: Sub-genotype. MENA: Middle East and North Africa.

**Table 2 pathogens-10-01333-t002:** The overall susceptibility/resistance to antiviral drugs among the MENA region HBV sequences.

Antiviral Drug ^1^	Trait	*n* ^4^ (%)
Lamivudine	S ^2^	2610 (95.5)
	R ^3^	124 (4.5)
Adefovir	S	2919 (99.2)
	R	23 (0.8)
Entecavir	S	2609 (96.2)
	R	103 (3.8)
Tenofovir	S	2941 (99.8)
	R	7 (0.2)
Telbivudine	S	2635 (95.7)
	R	117 (4.3)

^1^ Sequences excluded from analysis were those with compensatory mutations (for lamivudine, *n* = 75; for entecavir and telbivudine, *n* = 67); those with mutations of unknown value (for lamivudine, *n* = 214; for adefovir, *n* = 81; for tenofovir, *n* = 75; for entecavir, *n* = 244; and telbivudine, *n* = 204); ^2^ S: Susceptible; ^3^ R: Resistant, which included sequences assigned to the following categories (resistant/partly resistant/limited susceptibility); ^4^
*n*: Number.

**Table 3 pathogens-10-01333-t003:** Susceptibility/resistance to antiviral drugs among the MENA region HBV sequences divided by region and time of collection.

Antiviral Drug	Trait	Region		*p* Value ^4^	Time of Sequence Collection		*p* Value
		Middle East	North Africa		2001–2010	2011–2019	
		*n* ^3^ (%)	*n* (%)		*n* (%)	*n* (%)	
Lamivudine	S ^1^	1956 (94.3)	654 (99.1)	<0.001	1712 (95.6)	898 (95.2)	0.666
	R ^2^	118 (5.7)	6 (0.9)		79 (4.4)	45 (4.8)	
Adefovir	S	2268 (99.2)	651 (99.2)	0.948	1858 (99.1)	1061 (99.3)	0.557
	R	18 (0.8)	5 (0.8)		16 (0.9)	7 (0.7)	
Entecavir	S	2136 (95.5)	473 (99.6)	<0.001	1636 (96.1)	973 (96.4)	0.630
	R	101 (4.5)	2 (0.4)		67 (3.9)	36 (3.6)	
Tenofovir	S	2285 (99.8)	656 (99.5)	0.192	1869 (99.6)	1072 (100.0)	0.045
	R	4 (0.2)	3 (0.5)		7 (0.4)	0	
Telbivudine	S	1978 (94.6)	657 (99.4)	<0.001	1724 (95.9)	911 (95.5)	0.628
	R	113 (5.4)	4 (0.6)		74 (4.1)	43 (4.5)	

^1^ S: Susceptible; ^2^ R: Resistant; ^3^ *n*: Number; ^4^ *p* value: calculated using chi-square test. HBV: Hepatitis B virus; MENA: Middle East and North Africa.

**Table 4 pathogens-10-01333-t004:** Reverse transcriptase gene mutations among hepatitis B virus molecular sequences collected in the MENA region conferring antiviral resistance.

Antiviral Drug	Mutation *n* ^1^ (%)
Lamivudine	204I (*n* = 39); 180M,204I (*n* = 15); 180M,204V (*n* = 15); 173L,180M,204V (*n* = 8); 204I,80I (*n* = 6); 180M (*n* = 5); 180M,204I,80I (*n* = 5); 173Q,180G,204S,181T (*n* = 4); 180M,204V,80V (*n* = 4); 180M,204I,80V (*n* = 3); 173L,180M,204I,181T (*n* = 2); 181T (*n* = 2); 204I,80V (*n* = 2); 173A,180F,204E,181T,80H (*n* = 1); 173I,180R,204A,181T (*n* = 1); 173L,180M (*n* = 1); 173L,180M,204I,80I (*n* = 1); 173L,180P,204G,181T (*n* = 1); 173P,180P,204I (*n* = 1); 173R,180R,204P,181V,80H (*n* = 1); 173R,180R,204S,181V,80H (*n* = 1); 173W,180D,204L,181T (*n* = 1); 180M,204V,181V (*n* = 1); 181T,181S (*n* = 1); 181V (*n* = 1); 204S (*n* = 1); 204V (*n* = 1)
Adefovir	181T (*n* = 8); 236T (*n* = 3); 181T,236A (*n* = 2); 181V (*n* = 2); 181E,236T (*n* = 1); 181I,236T (*n* = 1); 181P,236T (*n* = 1); 181T,181S (*n* = 1); 181T,236L (*n* = 1); 181T,236T (*n* = 1); 181V,236Q (*n* = 1); 181V,236V (*n* = 1)
Entecavir	204I (*n* = 47); 204V,180M (*n* = 25); 204I,180M (*n* = 24); 184A,204V,180M (*n* = 1); 184S,204I,180M (*n* = 1); 202I,169G,184W,250C,204I,180P (*n* = 1); 202I,204I,180M (*n* = 1); 202I,204V,180M (*n* = 1); 204V (*n* = 1); 204V,202C,180M (*n* = 1)
Tenofovir	236T (*n* = 7)
Telbivudine	204I (*n* = 55); 204V (*n* = 24); 204I,80I (*n* = 12); 204I,80V (*n* = 5); 204R,181T (*n* = 4); 204V,80V (*n* = 4); 181T (*n* = 2); 204I,181T (*n* = 2); 181T,181S (*n* = 1); 181V (*n* = 1); 204A,181T (*n* = 1); 204E,80H,181T (*n* = 1); 204G,181T (*n* = 1); 204L,181T (*n* = 1); 204P,80H,181V (*n* = 1); 204S,80H,181V (*n* = 1); 204V,181V (*n* = 1)

^1^ *n*: Number; MENA: Middle East and North Africa; HBV: Hepatitis B virus.

**Table 5 pathogens-10-01333-t005:** Detailed description of the HBV phylogenetic clusters detected in the MENA region stratified by cluster size.

Genotype/SGT ^1^	Number of the MENA Sequences Analyzed ^2^	Number of Sequences within Clusters (%)	Dyads ^3^	Small Networks ^4^	Large Networks ^5^
D1	1777	267 (15.0)	Iran (31), Syria (3), Tunisia (1), Turkey (1), Iran and Oman (1)	Size 3: Iran (4), KSA (1), Tunisia (1), Egypt and KSA (1), Iran, Sudan and Syria (1); Size 4: Iran (7), Iran and Turkey (2), Iran and Sudan (1), Iran, Lebanon and Oman (1); Size 5: Iran (2); Size 6: Iran (2); Size 7: Iran (2); Size 8: Iran, Morocco, Lebanon, Turkey and Tunisia; Size 11: Iran (1); Size 12: Iran (1); Size 13: Iran and Turkey (1); Size 14: Iran (2)	Size 31: Iran, Oman, Palestine, Sudan, Tunisia, Turkey (1)
D7	268	166 (61.9)	Tunisia (4), Morocco and Tunisia (3), Somalia (2), Sudan and Tunisia (1)	Size 3: Morocco, Oman and Tunisia (1), Oman, Somalia and Tunisia (1), Tunisia (1); Size 4: Tunisia (2), Morocco and Tunisia (1); Size 6: Somalia and Tunisia; Size 7: Algeria, Morocco, Oman, Sudan and Tunisia (1), Morocco, Oman and Tunisia (1); Size 8: Morocco, Oman and Tunisia (1), Morocco, Sudan and Tunisia (1); Size 9: Morocco, Somalia and Tunisia (1), Oman (1), Oman, Sudan and Tunisia (1); Size 13: Algeria, Morocco, Sudan and Tunisia (1)	Size 15: Morocco and Tunisia (1), Morocco and Tunisia (1); Size 17: Morocco, Sudan and Tunisia (1); Size 31: Iran and Oman (1)
A1	102	0			
D2	78	22 (28.2)	Iran (4), Sudan (1), Syria (1)	Size 5: Iran (1), Iran and Lebanon (1)	
E	73	18 (24.7)	Sudan (1)	Size 4: Egypt and Sudan (1); Size 14: Sudan (1)	
D3	48	4 (8.3)		Size 4: Iran (1)	
A2	23	0			

^1^ SGT: Sub-genotype; ^2^ MENA: Middle East and North Africa; ^3^ Dyad: phylogenetic cluster with two sequences; ^4^ Small networks: phylogenetic clusters with three to 14 sequences; ^5^ Large networks: phylogenetic clusters with more than 14 sequences.

## Data Availability

The complete list of molecular sequences analyzed in this study can be found in GenBank (https://www.ncbi.nlm.nih.gov/genbank/ accessed on 30 April 2021) and by contacting the original submitting authors.

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
