# Peer review of "Variable Proportions of Phylogenetic Clustering and Low Levels of Antiviral Drug Resistance among the Major HBV Sub-Genotypes in the Middle East and North Africa"

_pathogens, 2021, doi:10.3390/pathogens10101333_

Round 1

Reviewer 1 Report

  1. It seems in the study the genotyping and phylogenetic clustering of HBV in MENA region were performed based on different sequence domains of the HBV genome. It would be clearer to put supplemental Fig.1 in the main text and depict which region of the HBV genome was selected for analysis of HBV genotypes or phylogenetic clustering. The research was undermined by using partial sequence but not full-length sequence, although the authors have discussed this limitation.
  2. In line 82 on page 2, the terms hyperendemic, intermediate endemicity, and low endemicity should be consistently used (noun, or adjective).
  3. In line 107, “The appears” should be corrected.
  4. Please explain why only the HBV sequences with the year of collection info (2375 sequences in total) were selected for antiviral drug resistance analysis. I recommended replacing “analysis” with “prediction”, given that there was no experimental evidence to confirm these predicted drug resistances. In table.2, the authors could consider adding one more row of “unknown” in addition to S and R.
  5. Table. 4 title was not right.
  6. Phylogenetic clustering is an important part of the study, the authors should consider generating a midpoint rooted phylogeographic tree (at least for genotype D) as the main figure, and please label phylogenetic clades on it. The conclusion drawn from the phylogenetic analysis was only “Transmission cluster phylogenetic analysis indicated the variable proportions of phylogenetic clustering”, not sufficiently investigated or clearly presented. The authors should’ve obtained more information about HBV dissemination across the different countries in MENA region.

Author Response

Reviewer #1 comments

  1. It seems in the study the genotyping and phylogenetic clustering of HBV in MENA region were performed based on different sequence domains of the HBV genome. It would be clearer to put supplemental Fig.1 in the main text and depict which region of the HBV genome was selected for analysis of HBV genotypes or phylogenetic clustering. The research was undermined by using partial sequence but not full-length sequence, although the authors have discussed this limitation.

Response: We would to thank the reviewer for this valuable comment. Based on the reviewer’s suggestion, we edited supplementary Figure 1, to include the region that was used to conduct maximum likelihood phylogenetic analysis for transmission cluster identification (GenBank accession NC_003977 positions 216-755), and added this Figure into the revised highlighted manuscript as Figure 7 (Please check page 11).

For genotyping, the Geno2pheno [hbv] 2.0 online tool uses any region of HBV even with variable sizes. Thus, we could not pinpoint every single region in the RT part of the genome used for genotyping since sequences were variable in size.

  1. In line 82 on page 2, the terms hyperendemic, intermediate endemicity, and low endemicity should be consistently used (noun, or adjective). 

Response: We would like to thank the reviewer for this comment, and accordingly we revised this sentence as follows: “countries can be classified as: countries with hyperendemicity (HBsAg prevalence of >8.0%), intermediate endemicity (HBsAg prevalence of 2.0–8.0%), and low endemicity with HBsAg prevalence <2.0%”. Line 82 of the revised highlighted manuscript.

  1. In line 107, “The appears” should be corrected 

Response: We would like to thank the reviewer for this comment, and accordingly we revised this sentence as follows: “This appears as an expected distribution”. Line 108 of the revised highlighted manuscript.

Additionally, we conducted a thorough check for minor English language and style mistakes.

  1. Please explain why only the HBV sequences with the year of collection info (2375 sequences in total) were selected for antiviral drug resistance analysis. I recommended replacing “analysis” with “prediction”, given that there was no experimental evidence to confirm these predicted drug resistances. In table.2, the authors could consider adding one more row of “unknown” in addition to S and R. 

Response: We would like to thank the reviewer for this important suggestion. The rationale behind selecting HBV sequences with the year of collection information was to evaluate whether certain patterns of temporal change in the prevalence of resistance could be detected (which was mentioned in the results, lines 265-266). Based on the reviewer’s valuable suggestion, we replaced the term “analysis” with the term “prediction” as follows: Line 250: “For antiviral drug resistance prediction”. For the suggestion of adding a row to include the mutations having “unknown” value, we prefer to keep the table in the current format considering that that details of HBV sequences falling under unknown category for each antiviral was included in the footnote of the table.

  1. Table. 4 title was not right. 

Response: We are deeply thankful for the reviewer for spotting this misplaced title. Accordingly, we corrected the title as follows: “Detailed description of the HBV phylogenetic clusters detected in the MENA region stratified by cluster size”

  1. Phylogenetic clustering is an important part of the study, the authors should consider generating a midpoint rooted phylogeographic tree (at least for genotype D) as the main figure, and please label phylogenetic clades on it. The conclusion drawn from the phylogenetic analysis was only “Transmission cluster phylogenetic analysis indicated the variable proportions of phylogenetic clustering”, not sufficiently investigated or clearly presented. The authors should’ve obtained more information about HBV dissemination across the different countries in MENA region. 

Response: We would like to thank the reviewer for this important comment. However, we have already provided the ML phylogenetic trees in the supplementary file since it would be hard to include each tree as a figure considering the number of sequences particularly for genotype D (SGT D1 and D7), with massive number of sequences which would be hard to follow if included in the main text.

For the comment on the conclusion that “Transmission cluster phylogenetic analysis indicated the variable proportions of phylogenetic clustering”, we disagree with the reviewer based on the results that were shown in Table 5, which indicated that clustering reached a percentage of 62% for SGT D7 compared to 28% for SGT D2, 25% for genotype E, 15% for SGT D1, and only 8% for SGT D3. Additionally, domestic phylogenetic clusters were not detected for SGT A1 and A2, indicating the wide variability in contribution to domestic HBV transmission for the major HBV genotypes/SGTs that circulate in the MENA region and justifies our conclusion. The detailed description of phylogenetic cluster analysis in the results with comments on inter-country spread of the virus is provided in section 3.5. of the results (lines 284-327). Thus, we prefer to keep the text in the current form.

Reviewer 2 Report

In introduction please add latest references for HBV from Polaris and WHO report, suggested below

  1. World Health Organization. 2021. Global progress report on HIV, viral hepatitis and sexually transmitted infections. Downloaded from: https://www.who.int/publications/i/item/9789240027077
  2. Polaris HBV Collaborators. Global prevalence, treatment and prevention of hepatitis b virus infection in 2016: a modelling study. Lancet Gastro & Hepato 2018, 3 (6): 383-403.

In discussion, please add a paragraph, with a couple of clinical studies on HBV in MENA region with different drugs and add the response of these drugs, if possible discuss the response of clinical hbv drug studies with the number of drug resistant mutants you got in this study.

I suggest minor revision

Author Response

Reviewer #2 comments

  1. In introduction please add latest references for HBV from Polaris and WHO report, suggested below

World Health Organization. 2021. Global progress report on HIV, viral hepatitis and sexually transmitted infections. Downloaded from: https://www.who.int/publications/i/item/9789240027077

Polaris HBV Collaborators. Global prevalence, treatment and prevention of hepatitis b virus infection in 2016: a modelling study. Lancet Gastro & Hepato 2018, 3 (6): 383-403.

Response: We would to thank the reviewer for this valuable comment, and based on the suggestions, we added the following references to the introduction section: “line 84, reference no. 25 (25. Polaris Observatory Collaborators. Global prevalence, treatment, and prevention of hepatitis B virus infection in 2016: a modelling study. Lancet Gastroenterol Hepatol 2018, 3,(6): 383-403, doi:10.1016/s2468-1253(18)30056-6.); and the following paragraph to the introduction section (lines 111-113) “Despite the recent reports indicating that the coverage of HBV vaccine has increased to exceed 80% for the third dose, the low birth dose coverage remains a challenge in the region” with the suggested reference no. 57 (57. World Health Organization (WHO). Global progress report on HIV, viral hepatitis and sexually transmitted infections. Available online: https://www.who.int/publications/i/item/9789240027077 (accessed on 10-10-2021)).

  1. In discussion, please add a paragraph, with a couple of clinical studies on HBV in MENA region with different drugs and add the response of these drugs, if possible discuss the response of clinical hbv drug studies with the number of drug resistant mutants you got in this study. 

Response: We would like to thank the reviewer for this suggestion, and accordingly we added the following paragraph: “The RT amino acid substitutions 204I/V were among the most frequent RT substitutions detected in the study. Such amino acid substitutions have been reported previously in Iran and Jordan, even among treatment-naïve individuals [50,116,117]. The amino acid substitutions 204I/V can be considered among the signature resistance mutations to lamivudine with cross-resistance to telbivudine [118,119]. The decreased response to lamivudine associated with selection of drug-resistant mutants has been reported in Egypt and Iran, and should be considered carefully, since this antiviral is considered a cost-effective treatment option widely used in the MENA region [71,120,121].”

The following references were included as well:

  1. Mahabadi, M.; Norouzi, M.; Alavian, S.M.; Samimirad, K.; Azad, T.M.; Saberfar, E.; Mahmoodi, M.; Ramezani, F.; Karimzadeh, H.; Malekzadeh, R., et al. Drug-related mutational patterns in hepatitis B virus (HBV) reverse transcriptase proteins from Iranian treatment-naïve chronic HBV patients. Hepatitis monthly 2013, 13,(1): e6712-e6712, doi:10.5812/hepatmon.6712.
  2. Amini-Bavil-Olyaee, S.; Hosseini, S.Y.; Sabahi, F.; Alavian, S.M. Hepatitis B virus (HBV) genotype and YMDD motif mutation profile among patients infected with HBV and untreated with lamivudine. Int J Infect Dis 2008, 12,(1): 83-87, doi:10.1016/j.ijid.2007.05.001.
  3. Lai, C.-L.; Dienstag, J.; Schiff, E.; Leung, N.W.Y.; Atkins, M.; Hunt, C.; Brown, N.; Woessner, M.; Boehme, R.; Condreay, L. Prevalence and Clinical Correlates of YMDD Variants during Lamivudine Therapy for Patients with Chronic Hepatitis B. Clinical Infectious Diseases 2003, 36,(6): 687-696, doi:10.1086/368083.
  4. Liu, Y.; Xu, Z.; Wang, Y.; Li, X.; Liu, L.; Chen, L.; Xin, S.; Xu, D. rtM204Q May Serve as a Novel Lamivudine-Resistance-Associated Mutation of Hepatitis B Virus. PLOS ONE 2014, 9,(2): e89015, doi:10.1371/journal.pone.0089015.
  5. Ismail, S.; Hafez, H.A.; Darweesh, S.K.; Kamal, K.H.; Esmat, G. Virologic response and breakthrough in chronic hepatitis B Egyptian patients receiving lamivudine therapy. Annals of gastroenterology 2014, 27,(4): 380-386.
  6. Ghandehari, F.; Pourazar, A.; Zadeh, M.S.; Tajedin, N. Probing rate of YMDD motif mutant in lamivudine treatment of Iranian patients with chronic hepatitis B virus infection. Asian journal of transfusion science 2011, 5,(1): 32-34, doi:10.4103/0973-6247.75982.